# Gay Community Stress Scale with Its Cultural Translation and Adaptions in Taiwan

**DOI:** 10.3390/ijerph191811649

**Published:** 2022-09-15

**Authors:** Chung-Ying Lin, Yu-Te Huang, Chiu-Hsiang Lee, Chia-Wei Fan, Cheng-Fang Yen

**Affiliations:** 1Institute of Allied Health Sciences, College of Medicine, National Cheng Kung University, Tainan 70101, Taiwan; 2Biostatistics Consulting Center, National Cheng Kung University Hospital, College of Medicine, National Cheng Kung University, Tainan 70101, Taiwan; 3Department of Public Health, College of Medicine, National Cheng Kung University, Tainan 70101, Taiwan; 4Department of Occupational Therapy, College of Medicine, National Cheng Kung University, Tainan 70101, Taiwan; 5Department of Social Work and Social Administration, The University of Hong Kong, Hong Kong RM543, China; 6Department of Nursing, Chung Shan Medical University, Taichung 40201, Taiwan; 7Department of Nursing, Chung Shan Medical University Hospital, Taichung 40201, Taiwan; 8Department of Occupational Therapy, AdventHealth University, Orlando, FL 32803, USA; 9Department of Psychiatry, Kaohsiung Medical University Hospital, Kaohsiung 80708, Taiwan; 10Department of Psychiatry, School of Medicine, Kaohsiung Medical University, Kaohsiung 80708, Taiwan; 11College of Professional Studies, National Pingtung University of Science and Technology, Pingtung 91201, Taiwan

**Keywords:** gay, bisexual men, stress, mental health, Taiwan

## Abstract

The present study aimed to adapt the Gay Community Stress Scale (GCSS) into the traditional Chinese version for measuring gay community stress experienced. Additionally, we examined its psychometric propensities among gay and bisexual men (GBM) in Taiwan. In total, 736 GBM participated in this study and completed the 35-item traditional Chinese version of the GCSS (29 items from the original GCSS and six items from the results of the focus group interviews among GBM in Taiwan); the Measure of Internalized Sexual Stigma for Lesbians and Gay Men (MISS-LG); the State-Trait Anxiety Inventory (STAI); and the Center for Epidemiological Studies Depression Scale (CES-D). Exploratory factor analysis results suggest a five-factor structure (i.e., Sex, Status, Competition, Exclusion, and Externals) for the 32-item traditional Chinese version of the GCSS among Taiwanese GBM; three items were deleted due to low factor loadings (i.e., <0.3). The five-factor “Externals” were not observed in the original GCSS. Moreover, the concurrent validity of the traditional Chinese version was supported by the positive correlations with MISS-LG, STAI, and CES-D. In conclusion, the traditional Chinese version of the GCSS showed relatively satisfactory psychometric properties. However, further research is needed to investigate the reasons for the possible etiology account for the different factor structures between the traditional Chinese version and the original GCSS.

## 1. Introduction

Gay and bisexual men (GBM) experience more severe mental health problems than their heterosexual peers [1,2]. According to minority stress theory [3,4], sexual stigma rooted in heterosexualism increase the risk of adverse health outcomes among GBM. Sexual stigma faced by GBM exists at multiple levels of daily life, including the individual level (such as internalized homonegativity [3]), the interpersonal level (such as homophobic bullying, perceived sexual stigma from the public, family and friends, and sexual orientation microaggression [4,5,6]), and the structural level (such as structured sexual stigma [7,8]).

Although evidence has supported the association between minority stressors due to heterosexist stigma and disparities in mental health in GBM [9,10], researchers continuously strive to develop newer models for explaining mental health disparities in GBM. Pachankis et al. [11] delineated intraminority gay community stress theory to illustrate how the gay community’s challenges impact the mental health of GBM. Contrary to sexual minority stressors originating from heterosexualism, intraminority gay community stress theory emphasizes that the mental health of GBM might be strongly determined by unique competitive pressures arising from the stress of social and sexual interactions within the gay community [11]. Intraminority gay community stress theory gained support based on the results of several previous studies. First, GBM often perceive and suffer from the rigid notions of status, including masculinity, attractiveness, and wealth, which are upheld by other GBM [12,13,14]. Second, compared to young heterosexual men, young GBM are more likely to derive their self-worth from achievement-related success through competition [15]. Third, sexual and social capital is distributed unequally in the gay community. For example, racial or ethnic minorities, older, or HIV-positive GBM often possess less sexual and social capital than others, with clear implications for their well-being [13,16,17,18,19].

Accordingly, Pachankis et al. [11] developed the Gay Community Stress Scale (GCSS) to measure the level of intraminority gay community stress among GBM. The original GCSS contains 29 items derived from qualitative interviews with GBM in the United States. An exploratory factor analysis (EFA) identified four factors comprising 20 items that described stress resulting from perceiving the gay community’s focus on sex, status, social competition, and the exclusion of diversity. Further confirmatory factor analyses confirmed the four-factor structure for the GCSS among GBM in two Western countries: the United States and Sweden [11]. The GCSS had satisfactory internal consistency and 1-year temporal stability. Perceived stress from the gay community measured by the GCSS predicted mental health and mediated the association between one’s gay community status and mental health [11]. 

Although the GCSS was developed using a robust procedure and has been approved as a reliable instrument for measuring intraminority gay community stress perceived by GBM [11], the developers emphasized the need for additional psychometric evidence for the GCSS. Specifically, Pachankis et al. [11] claimed that their study results examining the psychometrics of the GCSS could not be generalized beyond the distinct national contexts, as gay community stress might be comprised of different stressors depending on the global region. Importantly, the initial items of the GCSS were derived from a sample of GBM living in the United States; the four-factor structure was confirmed in the Western samples of GBM from the United States and Sweden [11]. Therefore, psychometric properties of the GCSS should be evaluated in other populations (e.g., Eastern countries such as Taiwan in the present study) to evaluate its generalizability beyond some specific groups such as White, educated, industrialized, rich, and democratic groups [20].

Asian societies have a lower tolerance of sexual minority individuals than Western societies [21]. A previous study found that Taiwan’s tolerance of sexual minority has outpaced China’s, Japan’s, and South Korea’s over the past two decades [21]. Liberal values related to divorce, prostitution, and gender roles have been considered mediators for the cohort improvement of tolerant attitudes toward sexual minorities in Taiwan [21]. However, research found that 38% and 32.6% of GBM reported themselves to be victims of sexuality-related bullying in the real and online world during childhood, respectively [22]. Sexuality-related bullying victimization in childhood significantly related to depression [23], anxiety [23], suicide [24], alcohol and illicit drug use [25,26], borderline personality symptoms [27], and problematic Internet use [28] in early adulthood. Moreover, research found that 26.5% of GBM reported having negative Facebook experiences due to their sexual orientation in the preceding year; sexuality-related negative Facebook experiences were significantly associated with an unsatisfactory quality of life [29]. In addition to bullying due to sexual orientation, microaggression [30] and internalized homonegativity [31] are also common among GBM in Taiwan; both were significantly associated with mental health problems [30,31]. Furthermore, some Taiwanese people have demonstrated discriminatory attitudes toward sexual minority individuals during the debate on legalizing same-sex relationships [32,33,34]. On 24 November 2018, people in Taiwan voted on same-sex marriage legalization; the result indicated that over 70% of voters opposed changing the Civil Code for the legalization of same-sex marriage [35]. In addition to sexual minority stressors, the role of intraminority gay community stress in mental health disparities warrants examination. Therefore, it is important to evaluate the experiences of sexual stigma and mental health problems among GBM in Taiwan. Translating and adapting the GCSS into a Taiwanese version according to its culture is the first step to understanding the intraminority gay community stress experienced by GBM living in Taiwan.

The present study aimed to examine whether the factor structure of the translated and culturally-adapted GCSS (i.e., a traditional Chinese version of GCSS) found among GBM living in Taiwan is the same as that found among GBM living in the United States. Moreover, given the socio-cultural background differences between Taiwanese and United States societies, we hypothesized that factor structure differences exist between the traditional Chinese version of the GCSS and the original GCSS.

## 2. Materials and Methods

### 2.1. Participants and Procedure

The participant inclusion criteria were men who identified their sexual orientation as gay or bisexual, were aged 20 or older, and were living in Taiwan. The participants were recruited via an advertisement on social media, including Facebook, Twitter, and LINE (a direct messaging app), the Bulletin Board System, and the home pages of three health promotion and counseling centers for sexual minority individuals from August 2021 to May 2022. Interested potential participants were asked to telephone the study’s research assistants, who ensured their eligibility, explained the study aims and procedures, and scheduled a time for them to complete the study questionnaires individually in the quiet study room. The research assistants evaluated the participants in the on-site study room to determine whether they had impaired intellect or showed signs of alcohol and substance use that might interfere with their understanding of the study’s purpose or completing the questionnaire. In total, 736 GBM participated in the study. No participant was excluded. Informed consent was obtained from all participants prior to the assessment. The study was approved by the Institutional Review Board of Kaohsiung Medical University Hospital (KMUHIRB-F(I)-20210003).

### 2.2. Measures

#### 2.2.1. Traditional Chinese Version of the Gay Community Stress Scale (GCSS)

Before beginning the formal research, we conducted three focus group interviews from March to May 2021 to help develop the traditional Chinese version of the CGSS, assessing the experiences of intraminority gay community stress among GBM living in the socio-cultural background of Taiwan. We recruited the focus group participants by posting an online advertisement on the home pages of three health promotion and counseling centers for sexual minority individuals. The recruitment criteria were GBM aged 20 or older who were born and lived in Taiwan most of the time and had engaged in gay community activities and social interactions for at least five years. A total of 24 GBM participated in the focus group interviews, with eight participants in each group. The participants’ mean age was 30.1 years (standard deviation [SD] = 3.4); 91.7% of participants identified as gay and 8.3% as bisexual; 87.5% of participants had completed college or university studies. The principal investigator led the group discussion on (1) the stress that the participants had experienced themselves or by other GBM in interacting with mainstream gay community members in Taiwan; (2) the stress experienced by GBM who wished to be mainstream gay community members in Taiwan. The principal investigator did not intentionally define the “mainstream gay community in Taiwan” to avoid limiting the participants from expressing their thoughts. Three researchers reviewed the transcript and coded the data for indications of gay community stressors. The principal investigator reviewed the coding results and compared them with the contents of the 29 items on the GCSS [11]. The results indicated that the coding of gay community stressors collected in the focus groups contained all 29 items on the GCSS or similar concepts.

In contrast, some concepts of gay community stressors collected in the focus groups did not appear in the 29 items on the GCSS. The principal investigator and three researchers discussed and formed six new items of gay community stressors in Taiwan in addition to the GCSS’s 29 items, including “The mainstream gay community views men who conceal sexual orientation from their family as losers” (item 30); “the mainstream gay community ignored their elders’ expectations for marrying and giving birth to babies” (item 31); “The mainstream gay community views men with a physical disability as less desirable” (item 32); “The mainstream gay community views men with chronic psychiatric illnesses as less desirable” (item 33); “The mainstream gay community views Taiwanese indigenous men as less desirable” (item 34); and “The mainstream gay community views men who live in nonurban regions as less desirable” (item 35). Thus, we used the 35-item traditional Chinese version of the GCSS for the exploratory factor analysis. The participants indicated the extent to which they perceived the item to be stressful (i.e., “How stressed/bothered are you by this potential aspect of the mainstream gay community?”) using a 5-point Likert scale from 1 (not at all stressed/bothered) to 5 (extremely stressed/bothered).

#### 2.2.2. External Criterion Measures

##### Measure of Internalized Sexual Stigma for Lesbians and Gay Men (MISS-LG)

The MISS-LG consists of 17 items embedded in three factors (i.e., Sexuality, Identity, and Social Discomfort). All the MISS-LG items were assessed using a 5-point Likert scale, where a score of 1 indicates *strongly disagree*, and a score of 5 indicates *strongly agree*. Therefore, a higher MISS-LG score indicates greater internalized sexual stigma [36]. Previous psychometric evidence showed that the MISS-LG was reliable and valid for the Taiwan population [37].

##### State-Trait Anxiety Inventory (STAI)

The STAI consists of 20 items embedded in a single factor of anxiety. All the STAI items were assessed using a 4-point Likert scale, where a score of 1 indicates *almost never* and a score of 4 *almost always*. Therefore, a higher STAI score indicates greater anxiety [38]. Previous psychometric evidence showed that the STAI was reliable and valid for the Taiwan population [39,40,41,42].

##### Center for Epidemiological Studies Depression (CES-D)

The CES-D consists of 20 items embedded in a single factor of depression. All the CES-D items were assessed using a 4-point Likert scale, where a score of 0 indicates *rarely or none of the time* and a score of 3 *most or all of the time*. Therefore, a higher CES-D score indicates greater depression [43]. Previous psychometric evidence showed that the CES-D was reliable and valid for the Taiwan population [44,45].

### 2.3. Statistical Analysis

The participants’ characteristics were first analyzed using frequency and percentage. Afterward, the GCSS items were analyzed using frequency, percentage, mean, SD, skewness, and kurtosis to illustrate their item properties. Skewness values < 3 together with kurtosis values < 10 indicate no serious problems in normal distribution for an item [46]. The factor structure of the CGSS was explored using EFA with an extraction method of principal axis factoring. Moreover, oblique rotation using the Oblimin method was applied for the EFA. A factor loading < 0.3 was considered low; items without a factor loading > 0.3 should be deleted [47]. When the EFA findings verified the factor structure of the CGSS, the internal consistency of each CGSS factor and its total score was calculated using Cronbach’s α, where a value > 0.7 indicates acceptability [48]. The concurrent validity of the CGSS (including its total score and factor scores) was examined using the Pearson correlation coefficients with the following external measures: three factors in the MISS-LG (i.e., Sexuality, Identity, and Social Discomfort), STAI, and CES-D. All the statistical analyses were performed using IBM SPSS 20.0 (IBM Corp., Armonk, NY, USA).

## 3. Results

Most of the participants (N = 736) were young adults (n [%] = 663 [90.1%]) and of Han ethnicity (n [%] = 728 [98.9%]). The majority of the participants had completed college or university studies (n [%] = 520 [70.7%]). Over four fifths of the participants (n [%] = 611 [83.0%]) were gay (Table 1).

Table 2 presents the GCSS item properties. More specifically, most of the items had higher proportions of the responses reported for *not at all stressed/bothered* (score 1) and *slightly stressed/bothered* (score 2). However, the values of skewness (range between 0.17 and 3.00) and those of kurtosis (range between −1.06 and 8.89) did not severely deviate from the normal distribution. Moreover, the mean scores of the GCSS items ranged between 1.28 and 2.81, indicating that the gay community stress level in the present sample was somewhat mild.

The EFA results indicated that GCSS items 1 to 6 (factor loadings = 0.580 to 0.870) embedded in the construct of Sex; items 7 to 10 with 17 (factor loadings = 0.386 to 0.938) in Status; items 11 to 16 (factor loadings = −0.845 to −0.515) in Competition; items 18 to 20 with 30 and 32 to 35 (factor loadings = 0.373 to 0.793) in Exclusion; items 21 to 24 with 26 and 28 to 29 (factor loadings = −0.804 to −0.435) in Externals. The construct of Externals is a new concept added to the original version of GCSS; the other constructs (i.e., Sex, Status, Competition, and Exclusion) are the same concepts proposed in the original version of GCSS. Additionally, it was suggested that items 25, 27, and 31 be deleted given that their factor loadings < 0.3 (Table 3). The final traditional Chinese version of the GCSS is shown in Appendix A.

The internal consistency values of the entire CGSS and all factors in the CGSS were satisfactory (α = 0.867 to 0.959). The inter-factor correlations in the CGSS were all in large effect sizes (r = 0.518 to 0.694; *p*-values < 0.001). Regarding the concurrent validity of the CGSS, it has significant correlations with the MISS-LG Sexuality factor (r = 0.136 to 0.263; *p*-values < 0.001); MISS-LG Identity factor (r = 0.187 to 0.309; *p*-values < 0.001); MISS-LG Social Discomfort factor (r = 0.144 to 0.278; *p*-values < 0.001); STAI score (r = 0.249 to 0.352; *p*-values < 0.001); and CES-D score (r = 0.303 to 0.421; *p*-values < 0.001) (Table 4).

## 4. Discussion

The present study’s EFA results indicated that the factor structure of the traditional Chinese version of the GCSS included five factors: Sex, Status, Competition, Exclusion, and Externals. The Sex, Status, and Competition factors of the GCSS’s traditional Chinese version had the same items as the 20-item version of the GCSS [11], except that item 17 was moved from the Competition factor of the GCSS’s 20-item version to the Status factor in the GCSS’s traditional Chinese version. In addition, five new items from this study’s focus group interviews were added to the Exclusion factor. Meanwhile, seven items that appeared in the original 29-item version but were removed from the 20-item version constructed a new factor of Externals in the traditional Chinese version of the GCSS.

The GCSS’s traditional Chinese version had almost the same items in the Sex, Status, and Competition factors as the original GCSS. The items in the Sex factor encompassed perceptions of the gay community’s hypersexuality and risky sex even at the expense of romantic relationships [11]. The Status factor contained the items regarding the gay community’s thinking highly of wealth and prestige [11]. The Competition factor contained the items describing fighting, gossip, and judgment within the gay community [11]. The results indicated that the gay community stressors contained in the Sex, Status, and Competition factors might commonly exist in both the Taiwan and United States gay communities and affect GBM across various socio-cultural backgrounds. The EFA results of the present study classified item 17 into the Status factor of the traditional Chinese version but not in the Competition factor of the 20-item version. Item 17 described “The mainstream gay community is overly materialistic.” Given that GBM who focus on materialism may expect wealth and prestige to meet their material needs, it is reasonable to classify item 17 in the Status factor of the traditional Chinese version of CGSS.

The original three items contained GBM’s perceived community exclusion based on racism, sexual racism, and HIV infection [11]. The EFA results of the present study expanded the contents of the Exclusion factor by adding five new items, including exclusion based on the concealment of sexual orientation from families, physical disability, psychiatric illnesses, indigenous identity, and nonurban residence. Sexual orientation concealment is a common strategy for GBM to protect against the stress of discrimination. However, it can simultaneously generate the stress of hiding and avoiding support from the gay community and significantly relates to mental health problems [49]. Sexual orientation concealment has been attributed to sexual stigma [50]; however, the present study demonstrated that concealing sexual orientation from family could also be an intraminority gay community stressor. Moreover, individuals with physical disabilities or mental illnesses are targets of public stigma [51].

The present study offered further evidence that physical disability and mental illness might increase GBM’s exclusion worries. The present study also found that GBM perceived that the community excluded indigenous GBM in Taiwan. Indigenous people account for 2.4% of the total population in Taiwan [52]. Acculturation [53] and health disparity [54] are plights encountered by Taiwanese indigenous people. The original GCSS contained the item “The mainstream gay community is racist.” The largest population of foreigners in Taiwan is the blue-collar migrant worker group from Southeast Asia. Stigma, isolation, and discrimination toward migrant workers have been serious social issues [55].

Given the different contexts of the gay community’s stress and the exclusion of Taiwanese indigenous people and migrant workers from Southeast Asia, we suggest evaluating both among GBM in Taiwan. Research has also evidenced that GBM living in rural regions are at a disadvantage concerning mental health and well-being compared with their urban counterparts [56,57]. The present study further supported that living in nonurban regions might be an intraminority community stressor for GBM in Taiwan.

The present study found that seven of the original version’s 29 items (21, 22, 23, 24, 26, 28 and 29) that were removed from the 20-item version constructed a new factor of Externals in the GCSS’s traditional Chinese version. The seven items contained GBM’s perceived community stress based on physically fit bodies, penis size, masculinity, age, social media, fitting into a specific category, and sexual position [11]. Most of the items on the Externals factor are related to sex and relational partner preference in GBM [58,59,60], indicating that partner preference in the mainstream gay community may contribute to intraminority stress in GBM. For example, item 26 describes “The mainstream gay community is overly preoccupied with social media.” Social media is an important source of social support for GBM [61]. In contrast, it can also be a space propagating the stereotyped social norm of outlooking and masculinity [62] and contribute to GBM’s gay community stress. Moreover, people in Taiwan are deeply influenced by Confucianism; people who grow up in the culture based on Confucianism are oriented more towards collectivism than those who grow up in Western culture [63]. The difference in value orientation may lead GBM in Taiwan to place greater importance on the stereotyped social norm of outlooking and masculinity than GBM in Western societies.

The results of the present study suggested deleting items 25 (“Within the mainstream gay community, strong, meaningful friendships are rare”); 27 (“In the mainstream gay community, there is a lot of drug use”); and 31 (“The mainstream gay community ignored their elders’ expectations for marrying and giving birth to babies”) from the GCSS’s traditional Chinese version. Although item 25 indicated the difficulty in maintaining friendships in the competitive gay community, it may result in less stress directly compared with the items of the Competition factor describing fighting, gossip, and judgment within the gay community. Drug use is a major health issue among GBM in Taiwan [64]; however, the present study did not support that it causes major stress for Taiwanese GBM. People in Taiwan traditionally emphasize the family obligations mandated in Confucianism to continue the family bloodline, whereas Taiwan’s mainstream gay community recommends that GBM be themselves and resist social pressure. Although Taiwan’s GBM may face this dilemma, this study did not support it as a major gay community stressor.

This study has several limitations. First, inherent social desirability biases in the questionnaires should be considered. Although we assured participants that all questionnaires and data would be anonymous and confidential, it is possible that the participants may have had other unvoiced concerns when they completed the questionnaires. Second, we did not test and retest the GCSS’s reliability and responsiveness; thus, it is unclear whether the traditional Chinese version has good reproducibility over time and is sensitive to detecting changes in gay community stress. Third, this study asked the participants to identify their gender as binary male or female; transgender, gender nonbinary, or genderqueer options were not offered. Research has found that sexual and gender minority identities have intersectional impacts on health [65] and behaviors [66]; both sexual and gender minority identities should be considered in public health practice [67].

## 5. Conclusions

This study’s results support the psychometric properties of the 32-item traditional Chinese version of the GCSS in a sample of GBM in Taiwan. The study also indicated that the traditional Chinese version of the GCSS shares several factors with the original GCSS, but also revealed differences in the overall factor structure between the traditional Chinese GCSS and the original GCSS. The results indicate that most of the gay community stressors might commonly exist in both the Taiwan and United States gay communities; however, the differences in some gay community stressors existed across various socio-cultural backgrounds. The traditional Chinese version of the GCSS can be applied to evaluate intraminority gay community stress and its influences on mental health, self-identity, and concepts of love and interpersonal relationships among GBM living in Taiwan.

## Figures and Tables

**Table 1 ijerph-19-11649-t001:** Participants’ characteristics (N = 736).

	n (%)
Age (year)	
20–29	335 (45.5)
30–39	328 (44.6)
40–49	65 (8.8)
50 or above	8 (1.1)
Ethnicity	
Han	728 (98.9)
Aborigine	8 (1.1)
Educational level	
Senior high or below	79 (10.7)
College or university	520 (70.7)
Postgraduate	137 (18.7)
Sexual orientation	
Gay	611 (83.0)
Bisexual	125 (17.0)

**Table 2 ijerph-19-11649-t002:** Item properties of the Gay Community Stress Scale (N = 736).

Item			n (%)			Mean	SD	Skewness	Kurtosis
	Score 1	Score 2	Score 3	Score 4	Score 5				
1	259 (35.2)	240 (32.6)	151 (20.5)	56 (7.6)	30 (4.1)	2.13	1.10	0.82	0.00
2	228 (31.0)	202 (27.4)	179 (24.3)	87 (11.8)	40 (5.4)	2.33	1.19	0.55	−0.62
3	267 (36.3)	208 (28.3)	149 (20.2)	75 (10.2)	37 (5.0)	2.19	1.18	0.73	−0.39
4	346 (47.0)	176 (23.9)	118 (16.0)	64 (8.7)	32 (4.3)	1.99	1.17	0.99	−0.02
5	304 (41.3)	167 (22.7)	125 (17.0)	88 (12.0)	52 (7.1)	2.21	1.29	0.75	−0.63
6	237 (32.2)	190 (25.8)	145 (19.7)	94 (12.8)	70 (9.5)	2.42	1.31	0.56	−0.83
7	331 (45.0)	163 (22.1)	150 (20.4)	57 (7.7)	35 (4.8)	2.05	1.18	0.88	−0.18
8	351 (47.7)	161 (21.9)	127 (17.3)	61 (8.3)	36 (4.9)	2.01	1.19	0.97	−0.08
9	385 (52.3)	158 (21.5)	105 (14.3)	58 (7.9)	30 (4.1)	1.90	1.16	1.14	0.28
10	299 (40.5)	173 (23.5)	151 (20.5)	79 (10.7)	34 (4.6)	2.15	1.20	0.73	−0.51
11	336 (45.7)	189 (25.7)	109 (14.8)	67 (9.1)	35 (4.8)	2.02	1.18	0.99	−0.02
12	304 (41.3)	199 (27.0)	120 (16.3)	70 (9.5)	43 (5.8)	2.12	1.21	0.89	−0.22
13	331 (45.0)	182 (24.7)	121 (16.4)	66 (9.0)	36 (4.9)	2.04	1.19	0.94	−0.11
14	317 (43.1)	187 (25.4)	119 (16.2)	56 (7.6)	57 (7.7)	2.12	1.26	0.95	−0.16
15	355 (48.2)	186 (25.3)	114 (15.5)	53 (7.2)	28 (3.8)	1.93	1.13	1.09	0.31
16	256 (34.8)	259 (35.2)	136 (18.5)	56 (7.6)	29 (3.9)	2.11	1.09	0.88	0.15
17	317 (43.1)	219 (29.8)	117 (15.9)	57 (7.7)	26 (3.5)	1.99	1.10	1.00	0.21
18	388 (52.7)	181 (24.6)	89 (12.1)	45 (6.1)	33 (4.5)	1.85	1.13	1.30	0.84
19	503 (68.3)	119 (16.2)	62 (8.4)	31 (4.2)	21 (2.9)	1.57	1.01	1.87	2.77
20	402 (54.6)	140 (19.0)	99 (13.5)	52 (7.1)	43 (5.8)	1.90	1.22	1.20	0.34
21	150 (20.4)	166 (22.6)	193 (26.2)	127 (17.3)	100 (13.6)	2.81	1.31	0.17	−1.06
22	276 (37.5)	154 (20.9)	137 (18.6)	100 (13.6)	69 (9.4)	2.36	1.35	0.57	−0.93
23	229 (31.1)	196 (26.6)	152 (20.7)	87 (11.8)	72 (9.8)	2.43	1.30	0.57	−0.78
24	336 (45.7)	180 (24.5)	109 (14.8)	64 (8.7)	47 (6.4)	2.06	1.24	0.99	−0.10
25	337 (45.8)	185 (25.1)	118 (16.0)	58 (7.9)	38 (5.2)	2.01	1.18	1.01	0.06
26	334 (45.4)	187 (25.4)	105 (14.3)	54 (7.3)	56 (7.6)	2.06	1.25	1.03	0.00
27	371 (50.4)	135 (18.3)	111 (15.1)	62 (8.4)	57 (7.7)	2.05	1.30	0.99	−0.24
28	231 (31.4)	221 (30.0)	120 (16.3)	84 (11.4)	80 (10.9)	2.40	1.32	0.66	−0.73
29	242 (32.9)	190 (25.8)	146 (19.8)	87 (11.8)	17 (9.6)	2.40	1.31	0.59	−0.78
30	498 (67.7)	104 (14.1)	64 (8.7)	37 (5.0)	33 (4.5)	1.65	1.12	1.73	1.97
31	335 (45.5)	157 (21.3)	118 (16.0)	68 (9.2)	58 (7.9)	2.13	1.30	0.89	−0.39
32	464 (63.0)	136 (18.5)	78 (10.6)	26 (3.5)	32 (4.3)	1.68	1.08	1.67	2.03
33	472 (64.1)	119 (16.2)	87 (11.8)	32 (4.3)	26 (3.5)	1.67	1.07	1.60	1.72
34	624 (84.8)	50 (6.8)	37 (5.0)	16 (2.2)	9 (1.2)	1.28	0.76	3.00	8.89
35	518 (70.4)	106 (14.4)	62 (8.4)	28 (3.8)	22 (3.0)	1.55	1.00	1.95	3.07

**Table 3 ijerph-19-11649-t003:** Results of factor loadings derived from exploratory factor analysis for the Gay Community Stress Scale (GCSS).

Item			Factor Loading			Embedded Factor
	Sex	Status	Competition	Exclusion	Externals	in Original GCSS
1	0.621	--	--	--	--	Sex
2	0.658	--	--	--	--	Sex
3	0.795	--	--	--	--	Sex
4	0.870	--	--	--	--	Sex
5	0.814	--	--	--	--	Sex
6	0.580	--	--	--	--	Sex
7	--	0.757	--	--	--	Status
8	--	0.938	--	--	--	Status
9	--	0.835	--	--	--	Status
10	--	0.386	--	--	--	Status
11	--	--	−0.805	--	--	Competition
12	--	--	−0.832	--	--	Competition
13	--	--	−0.845	--	--	Competition
14	--	--	−0.842	--	--	Competition
15	--	--	−0.515	--	--	Competition
16	--	--	−0.531	--	--	Competition
17	--	0.407	--	--	--	Competition
18	--	--	--	0.426	--	Exclusion
19	--	--	--	0.534	--	Exclusion
20	--	--	--	0.456	--	Exclusion
21	--	--	--	--	−0.804	Not included
22	--	--	--	--	−0.662	Not included
23	--	--	--	--	−0.737	Not included
24	--	--	--	--	−0.459	Not included
25	--	--	--	--	--	Not included
26	--	--	--	--	−0.435	Not included
27	--	--	--	--	--	Not included
28	--	--	--	--	−0.562	Not included
29	--	--	--	--	−0.470	Not included
30	--	--	--	0.373	--	--
31	--	--	--	--	--	--
32	--	--	--	0.710	--	--
33	--	--	--	0.793	--	--
34	--	--	--	0.770	--	--
35	--	--	--	0.588	--	--

Note. Factor loadings < 0.35 were not reported; items 30 to 35 were Taiwan cultural items developed by the present authors and were not in the original GCSS.

**Table 4 ijerph-19-11649-t004:** Concurrent validity and internal consistency of the Gay Community Stress Scale (GCSS).

External Measure			GCSS			
	Sex (α = 0.889)	Status (α = 0.900)	Competition (α = 0.924)	Exclusion (α = 0.867)	Externals (α = 0.899)	Total Scale (α = 0.959)
GCSS						
Sex	--					
Status	0.544	--				
Competition	0.597	0.694	--			
Exclusion	0.518	0.611	0.639	--		
Externals	0.616	0.651	0.654	0.661	--	
Total score	0.784	0.824	0.857	0.825	0.874	--
MISS-LG						
Sexuality	0.263	0.136	0.194	0.139	0.228	0.233
Identity	0.309	0.207	0.237	0.187	0.212	0.275
Social Discomfort	0.278	0.144	0.184	0.212	0.237	0.256
STAI	0.280	0.349	0.273	0.249	0.323	0.352
CES-D	0.303	0.395	0.366	0.307	0.388	0.421

Note. All *p*-values < 0.001 for the Pearson correlation coefficients. CES-D = Center for Epidemiological Studies Depression; GCSS = Gay Community Stress Scale; MISS-LG = Measure of Internalized Sexual Stigma for Lesbians and Gay Men; STAI = State-Trait Anxiety Inventory.

## Data Availability

The data used in this study are available upon reasonable request to the corresponding authors.

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
