# Peer review of "Gay Community Stress Scale with Its Cultural Translation and Adaptions in Taiwan"

_ijerph, 2022, doi:10.3390/ijerph191811649_

Round 1
Reviewer 1 Report
The article is based on a large research sample (736 gay and bisexual men in Taiwan). It is a valuable contribution to research into the situation of sexual minorities in Asia. The part concerning the situation of sexual minorities in Taiwan is very short, and even laconic (pp. 2-3). In my opinion, this part should be worked out in order to indicate what stressors may occur in the case of LGBT people in the specific conditions of the Taiwanese society. The research shows some interesting results on the gay male community in Taiwan (e.g. “The present study also found that GBM perceived the community excluded indigenous GBM in Taiwan”, p. 8). In my view, the conclusions section (p. 9) is too sketchy.
Author Response
Comment 1
The article is based on a large research sample (736 gay and bisexual men in Taiwan). It is a valuable contribution to research into the situation of sexual minorities in Asia. …The research shows some interesting results on the gay male community in Taiwan (e.g. “The present study also found that GBM perceived the community excluded indigenous GBM in Taiwan”, p. 8).
Response
Thank you for your support.
Comment 2
The part concerning the situation of sexual minorities in Taiwan is very short, and even laconic (pp. 2-3). In my opinion, this part should be worked out in order to indicate what stressors may occur in the case of LGBT people in the specific conditions of the Taiwanese society.
Response
We expanded the introduction for the sexual minority stressors among LGBT people in Taiwan as below. Please refer to line 92-110.
“A previous study found that Taiwan’s tolerance of sexual minority has outpaced China’s, Japan’s, and South Korea’s over the past two decades [21]. Liberal values related to divorce, prostitution, and gender roles have been considered mediators for cohort improvement of tolerant attitudes toward sexual minority in Taiwan [21]. However, research found that 38% and 32.6% of GBM reported themselves to be victims of sexuality-related bullying in real and cyber worlds during childhood, respectively [22]. Sexuality-related bullying victimization in childhood significantly related to depression [23], anxiety [23], suicide [24], alcohol and illicit drug use [25,26], borderline personality symptoms [27], and problematic Internet use [28] in early adulthood. Moreover, research found that 26.5% of GBM reported having negative Facebook experiences due to their sexual orientation in the preceding year; sexuality-related negative Facebook experiences were significantly associated with unsatisfactory quality of life [29]. In addition to bullying due to sexual orientation, microaggression [30] and internalized homonegativity [31] are also common among GBM in Taiwan; both were significantly associated with mental health problems [30,31]. Furthermore, some Taiwanese people have demonstrated discriminatory attitudes toward sexual minority individuals during the debate on legalizing same-sex relationships [32-34]. On November 24, 2018, people in Taiwan voted on same-sex marriage legalization; the result indicated that over 70% of voters opposed changing the Civil Code for the legalization of same-sex marriage [35].”
Comment 3
In my view, the conclusions section (p. 9) is too sketchy.
Response
Thank you for your comment. We expanded the conclusion section as below. Please refer to line 345-354.
“This study’s results supported the psychometric properties of the 32-item traditional Chinese version of the GCSS in a sample of GBM in Taiwan. The study also indicated that the traditional Chinese version of the GCSS shared several factors with the original GCSS but also revealed differences in the overall factor structure between the traditional Chinese GCSS and the original GCSS. The results indicated that most of the gay community stressors might commonly exist in both the Taiwan and United States gay communities; however, the differences in some gay community stressors existed across various socio-cultural backgrounds. The traditional Chinese version of the GCSS can be applied to evaluate the intraminority gay community stress and its influences on mental health, self-identity, and concepts of love and interpersonal relationship among GBM living in Taiwan.”
Reviewer 2 Report
This is a timely and rigorous study that seeks to validate and adapt the Gay Community Stress Scale to the Taiwan context. The key merits of this study include the use of focus groups to identify important themes related to the gay community in Taiwan, a comprehensive psychometric evaluation of the adapted/expanded GCSS, and a meaningful and informative discussion of how the adapted GCSS was similar or different from the original scale. The writing and methods are both of high quality. As such I do not have any major comments.
Minor issues:
1. In Table 3, the "Deleted" notation for the externals factor items was confusing. The items that were not included in the original GCSS should be better differentiated from items that did not have high enough factor loadings to be retained in the current study.
2. The discussion is very well written. I wonder if some of the findings can be further improved by contextualizing differences between eastern and western cultures in terms of their respective emphasis on collectivism and individualism, particularly on items that are added in the Taiwanese version of the GCSS.
3. Can the authors provide in an appendix with the final items of the adapted GCSS and scoring information for the new scale in both Traditional Chinese and in English?
4. A small typo on lines 318-320, where "It" should be "it"
Author Response
Comment
This is a timely and rigorous study that seeks to validate and adapt the Gay Community Stress Scale to the Taiwan context. The key merits of this study include the use of focus groups to identify important themes related to the gay community in Taiwan, a comprehensive psychometric evaluation of the adapted/expanded GCSS, and a meaningful and informative discussion of how the adapted GCSS was similar or different from the original scale. The writing and methods are both of high quality. As such I do not have any major comments.
Response
Thank you for your support.
Minor issues:
Comment
- In Table 3, the "Deleted" notation for the externals factor items was confusing. The items that were not included in the original GCSS should be better differentiated from items that did not have high enough factor loadings to be retained in the current study.
Response
Thank you for your comment. We replaced “Deleted” by “Not included” in Table 3 to indicate items that were not included in the original GCSS.
Comment
- The discussion is very well written. I wonder if some of the findings can be further improved by contextualizing differences between eastern and western cultures in terms of their respective emphasis on collectivism and individualism, particularly on items that are added in the Taiwanese version of the GCSS.
Response
Thank you for your suggestion. We added the discussion regarding the role of collectivism-individualism in items that are added in the Taiwanese version of the GCSS. Please refer to line 313-318
“Moreover, people in Taiwan are deeply influenced by the Confucianism; people who grow up in the culture based on Confucianism is more collectivistic-orientated than those who grow up in Western culture [63]. The difference in value orientation may make GBM in Taiwan place greater importance on the stereotyped social norm of outlooking and masculinity than GBM in Western societies.”
Comment
- Can the authors provide in an appendix with the final items of the adapted GCSS and scoring information for the new scale in both Traditional Chinese and in English?
Response
We added the final items of the Traditional Chinese version of the GCSS and scoring information into the revised manuscript as an appendix. Please refer to line 537-546. However, we did not include the English version of the GCSS because of copyright.
Comment
- A small typo on lines 318-320, where "It" should be "it"
Response
Thank you for your reminding. We corrected it. Please refer to line 335.